# An Analysis of Post-Apartheid Anti-Fronting Interventions Fostering Mainstreaming of the Black South Africans into Corporate Sector

Treasure Hlayisani Mathebula and Kolawole Olusola Odeku *

Department of Public and Environmental Law, University of Limpopo, Polokwane 0727, South Africa; 200911245@keyaka.ul.ac.za
* Correspondence: kolawole.odeku@ul.ac.za

**Abstract:** While the colonial and apartheid regimes utilised draconian, arbitrary, segregated, discriminatory, and exclusive anti-black social-socioeconomic policies and laws to deny the majority of black South Africans access to and participation in various economic activities, post 1994 democratic South Africa has strategically introduced progressive policies and laws to ensure that black South Africans play active productive roles in socio-economic activities in all sectors. While this is commendable, various corporations and businesses owned by white companies are supposed to ensure that black people become part and parcel of the businesses, and companies are being denied active participation, using fronting purposefully to circumvent the requirements of the anti-fronting laws. Against this backdrop, this paper seeks to analyse the post-apartheid anti-fronting interventions that foster the mainstreaming of black South Africans into the corporate sector. The paper uses a literature review methodology to find and analyse primary and secondary sources of data relating to equality, BEE, and fronting. This paper presents the historical exclusion of blacks through the instrumentality of colonial and apartheid apparatuses and laws brutally utilised to exclude blacks from economic activities. Post 1994 democratic transformative interventions—laws—have been enacted to redress the segregated and exclusive laws; however, fronting activities and practices continue to undermine and circumvent successful implementation. This said, the post 1994 government continues to tackle impunity through the exploration of civil and criminal responsibilities and accountability of perpetrators and use the rule of law-judicial means to hold perpetrators accountable. This paper found that fronting is a persisting issue in South Africa despite anti-fronting legislative measures developed over the past years when the B-BBEE Act was amended. This paper advises more on pro-active anti-fronting measures to pro-actively foster the mainstreaming of black South Africans into the corporate sector.

**Keywords:** social-socioeconomic interventions; corporate sector; inclusivity; fronting; black South Africans

## 1. Introduction

The historical background of the gradual development or evolution of the economy in South Africa has over the years been reflected or based on the marginalisation, exploitation, and exclusion of black people from all social and socioeconomic activities in the country. Colonialism and apartheid were used to keep the white minority in power, and as a result, wealth and income distribution in South Africa were unequal. The whites live in affluence, while the black majority are permanently in deep poverty. Land dispossessions of black people and the institutionalisation of the system that controlled the economy and employed black people as cheap labour were commonplace prior to the democratic dispensation in 1994 (Morifi and Mahlatsi 2021, p. 277). The colonial and apartheid systems of governance promulgated laws that restricted black people from buying and owning land that was valuable for mining and agriculture (Strauss 2019). Gender and racial oppression were key features of the apartheid system, and it resulted in widespread maltreatment of women,

mainly black women and black people, reflected in the type and quality of education and economic opportunities they could access (Gallo 2020, p. 6). The apartheid system offered poor education to black people, which limited their opportunities. The black people's kind of education and economic participation were, of course, not enough to allow them to compete in the mainstream of education; hence, quality jobs and education were reserved only for the white people (Gallo 2020, p. 7). The colonial and apartheid systems and policies have given rise to vast economic and systematic contortions that led to serious economic catastrophes and continued beyond that. The period between 1970 and 1994 was confronted by challenges such as poverty, inequality, sluggish economic growth, surging unemployment, a lack of investments, a lack of education, and a lack of skills for a poorer, higher cost of living (Ally and Lissoni 2017, p. 34). Post-apartheid, the black majority inherited a mismanaged economy and continues to fix the damages done during the colonial and apartheid eras to date (Polus et al. 2020, p. 296). The new dispensation seeks to eradicate poverty, inequality, racial discrimination, and unemployment (Shezi 2021, p. 69). Despite this, the World Bank and International Monetary Fund (2020) reported that South Africa is the most unequal country in the world, with race playing a determining factor in a society where the minority population owns more than 80 percent of the wealth.

It is pertinent to point out that South Africa is now a constitutional democracy, and the current government is led by the black majority, the ruling African National Congress (ANC). This feat has established greater political equality, but attainment of political power and equality alone could not be enough to address inherited social and economic inequities, hence the need for economic equality, which will have ripple effects and change the social conditions of the black majority. Essentially, the ANC government's policies, such as the Ready to Govern as well as the Reconstruction and Development Programme (RDP), are initial post-apartheid policies introduced to foster social and socioeconomic emancipation and alleviate social ills and poverty. Other social and socioeconomic policy interventions to strengthen the deployment and delivery of social and socioeconomic goods and amenities to the people were the Growth, Employment, and Redistribution (GEAR) of 1996, the African Accelerated and Shared Growth Initiative (AsgiSA) in 2006, the New Growth Path (NGP) in 2010, and the National Development Plan (NDP) in 2012 (Gelb 2006, p. 4). In the same vein, the government strategically legislates Black Economic Empowerment to enable blacks to participate and be owners of companies in South Africa. Generally, the Broad-Based Black Economic Empowerment Strategy Document of 2003 (B-BBEE Strategic Document) is a very crucial intervention or assistance by the government to remedy the majority of South Africans' persistent exclusion from full economic participation or engagement. It consists of three main components: firstly, absolute empowerment by ownership, including control of businesses and their assets: secondly, human resource development and parity in the workplace; and thirdly, indirect emancipation by preferential procurement, including the development of businesses. According to the Trade and Industry Department (DTI), the B-BBEE strategic document provides for the state to spearhead the process of development and ensure the implementation or execution of the cogent and consolidated strategy to accomplish broader goals of economic empowerment as well as ensure that there is equal engagement in economic activities. The guiding principles of the economic reforms encompass certain crucial aspects. The first aspect advocates for a broad-based approach as a means to address the potential for social and political instability caused by income inequities based on race. The second aspect champions inclusiveness, as a fairer economy would benefit all individuals in South Africa, irrespective of their backgrounds. The third aspect notes that efficient governance plays a crucial role in ensuring that economic reform programmes and economic transformation strategies adhere to high standards of transparency and accountability. The last aspect avers that the BEE strategy should form part of a larger strategy for growth, which includes the growth of the socio-economic system, all development programmes, and the BEE all working together.

While this legislative intervention was applauded, some white-owned companies in cohort with some black people have continually used fronting to circumvent and defeat

the purpose, aims, and objectives of empowering black people by mainstreaming them in existing or new companies (du Plessis and BProc 2022, p. 398). The on-going problem causing challenges to achieving substantive equality and inclusion of black people in mainstream economic activities in South Africa is the growing rate of fronting practices (Department of Trade, Industry, and Competition 2020).

According to the Department of Trade, Industry, and Competition, "fronting means a deliberate circumvention or attempted circumvention of the B-BBEE Act and the Codes. Fronting commonly involves reliance on data or claims of compliance based on misrepresentations of facts, whether made by the party claiming compliance or by any other person." In summary, it entails situations where businesses use black people to front, pretending to be compliant with the B-BBEE interventions as if they are rear. Gerber and Curlewis (2018, p. 372) articulated that fronting practices transpire when there is an inclusion of historically disadvantaged people within mainstream economic activities or as shareholders but without an actual transfer of wealth or management, which is consistent with the B-BBEE Act of 2013. In addition, Matotoka and Odeku (2022, p. 3) posit that persons used as frontiers in fronting practices do not perform the actual management duties nor do they enjoy the actual benefits from economic activities of corporations as people in their positions ordinarily receive, but they are merely appointed as frontiers for the sole purpose of being B-BBEE compliant. According to Sibanda (2015, p. 31), companies that are perpetrators of fronting offences usually misrepresent information in order to obtain the B-BBEE certificate and use such a certificate to accumulate benefits that they would not have gotten had they not misrepresented the information. Sibanda (2015, p. 30) further noted that, taking into account the immediate and long-term implications of fronting practices, companies committing fronting offences negatively affect key elements of corporate governance.

Whereas this is a form of window dressing where black employees appear to be part of the company as directors and shareholders in a real sense, this is pure misrepresentation. To accomplish fronting, the rank and file in a company, like a driver or gardener, are projected as directors in order for the company to appear to meet the B-BBEE standard and secure lucrative tender businesses and jobs. Fronting clouds the government's ability to track the record of B-BBEE progress towards achieving the B-BBEE objectives and renders the need to develop effective anti-fronting mechanisms inevitable. Against this background, this paper uses literature review methodology by analysing primary and secondary sources of law to appraise the post-apartheid anti-fronting interventions fostering the mainstreaming of black South Africans into the corporate sector.

## 2. Methodology

The study is grounded in the literature review method with which the primary and secondary sources of law were collected and analysed in relation to the title of this paper. This paper thus analyses different sources, such as the Constitution, statutes, case law, policies, and other primary sources of law. In addition, the author uses journal articles, books, internet articles, and theses. As this exclusively pays attention to the B-BBEE strategy to foster inclusion of black people in mainstream economic participation, the B-BBEE Act (with its amendments), case law, and the constitution were the most consulted sources. In addition, review was conducted to analyse what other authors have published on the areas of the impact of the BEE policy objectives and the impediment of the policy due to BEE fronting practices.

## 3. The Black Economic Empowerment Commission 2001

In order to implement the BEEE, the Black Economic Empowerment Commission (BEE Commission) was established in 1998 following a proposal by the Black Economic Forum, which represented at least eleven (11) organisations of black businesses with the aim and objectives of the following:

- To conduct an empirical study on the BEE process and monitor the rate and outcomes of BEE activities during the period 1990.

- Make any findings or conclusions on the barriers to black people's meaningful economic engagement or participation.
- come up with a compelling case scenario for an expedited national BEE approach and make policy and instrument suggestions to lead a long-term strategy.
- To provide benchmarks and criteria for monitoring the national BEE strategy's execution.

The BEE Commission embarked on substantial research in order to address the main objectives as set out and held various consultations with stakeholders. The BEE Commission, in its report, made recommendations to address underdevelopment as well as the meaningful participation of black people in the economic mainstream to address the lingering legacy of colonial and apartheid systems of oppression and disempowerment (Modise and Mtshiselwa 2013, p. 362). As a result, the BEE Commission proposed that an "Integrated National BEE Strategy" be adopted, consisting of a synchronised, harmonised, less complicated, and modernised process. This involves a collection of rules and laws that establish defined aims and demarcate duties and obligations for the corporate sector, governmental sector, and civil society.

Since its release, the BEE Commission Report has played a critical role in the development of the program. In short, the report advocated for the state to have a far more active and interventionist role in encouraging and pursuing black empowerment. To this end, the BEE Commission made recommendations that the government enact laws to govern the empowerment of the black majority. Hence, the BEE Act was promulgated. The Act, among other things, intends to assist the government to identify and formalise procedures, targets, and other measures with which economic actors can be better informed about BEE achievements, "finance specific investment projects, and formalise procedures, indicators, targets, and other measures with which economic actors can be better informed about BEE accomplishments. The BEE Commission concluded that the apartheid government monumentally failed to provide the necessary or sufficient institutional and financial assistance for long-term black empowerment, as well as having targeted the people based on racial considerations. The panel also recommended the establishment of regulatory bodies that shall discharge the BEE duties, such as a procurement agency and public financial companies like the National Empowerment Funding Agency (NEFA), to centralise and focus policy action and procedures.

## 4. The Broad-Based Black Economic Empowerment Act 53 of 2003 (BBBEE, 2003)

To address the social-socioeconomic imbalances established by Apartheid policies that privileged white business owners, the post-apartheid government decided that active action in asset and opportunity redistribution was essential (Lahiff et al. 2007, p. 1418). The B-BBEE Act was enacted purposefully "to transform the economy to be representative of the demographics, specifically the race demographics of the country. The B-BBEE Act was passed in 2003 with the intent of creating a framework for promoting black economic development through legislation, granting the minister such powers to publicise codes of good conduct and charters or directives related to transformation, and establishing a council that serves as an advisory body on BEE-related matters. The Act's main goals are to advance the constitutional right to equality, increase the broad-based and effective participation of black people in the economy, promote a higher growth rate, increase employment, and ensure more equitable income distribution, as well as establish a national policy on broad-based economic empowerment to promote the nation's economic unity and equal opportunity and access to government services (Amoah 2023, p. 16).

The statute uses the term "black people" to refer to Africans, coloureds, and Indians. Black people—the combination of those that have been mentioned and should have an effect on women, youth, workers, people with disabilities, and all those that were previously disadvantaged due to their race and are usually based in rural areas with fewer opportunities—can benefit from the law. Increasing proportions of black persons in management positions who own and manage businesses, including assets, initiate community projects, employees, cooperatives, and ownership and administration of firms

and economic assets by other collective enterprises, human capital, empowerment and development of skills, obtaining equal representation in all fields and workplace positions, and preferential procurement and investment in black-owned businesses are all examples of empowerment. However, the setback is that, since the promulgation of the Act, several businesses have resorted to deceptive techniques (such as fronting) in order to improve their BEE score. Certain businesses and white persons have taken advantage of the Act by appointing secretaries, tea ladies, and gardeners as directors, frequently without their knowledge, in order to circumvent the law and qualify for government tenders (Van de Rheede 2020, p. 104). In order to address this fronting problem, the government introduced an amendment, the "Broad-Based Black Economic Empowerment Amendment Act (B-BBEE Act 46 of 2013). Sadly, fronters continue to poke holes in the prescripts of the law and continue to circumvent the law. For example, the Department of Trade, Industry, and Competition (2020) received more than 680 cases of fronting reports, and the department stated that the rising incidents of fronting place pressure on the success of the B-BBEE Act. However, there is a shortage of published data, which determines the extent to which fronting is perpetrated in South Africa. The Department of Trade, Industry, and Competition is responsible for capturing fronting incidents and reports. Fronting practices are often traced through case law. Even though there are diverse opinions on the anti-fronting laws, it is apt to point out that the government interventions bode well, as the objective was to mainstream the previously denied and disadvantaged black majority to participate and contribute to the social and socio economics activities of the country. This said, we shall now look at the nitty-gritty of all the concerns of the pundits, whether for or against.

## 5. Transformative Constitutionalism, Rights, and Equity

The enactment of the Constitution became the biggest and most progressive step towards eradicating unfair racial discrimination stemming from the apartheid regime. Section 9 of the Constitution seeks to achieve substantive equality. Substantive equality gave rise to the transformative constitutionalism mandate, which seeks to achieve equality. Looking at it from the perspective of a value and as a right (Albertyn and Goldblatt 1998, p. 255) indicates that the right to equality is pivotal to achieving socio-economic transformation in South Africa, and given the historical background, this will necessitate prioritising certain categories. Thus, some groups, such as white people, mostly males, are fairly discriminated against in order to transform the lives of black people generally, and mostly black women in particular.

The drafters of the Constitution took into consideration international law and South Africa's historical socio-economic inequalities, mainly due to apartheid, when incorporating the equality clause (Section 9) as a constitutional fundamental right (Rapatsa 2015, p. 209). Consequently, it can be safely said that the equality clause in the Constitution should be given more interpretation of substantive equality than formal equality, although the clause envisages both types of equalities. Equality is explicitly provided for in Section 9 of the Constitution, which states that everyone is equal before the law and should never be discriminated against unfairly on the grounds of, but not limited to, gender, race, sexual orientation, age, background, culture, etc. The scholars have over the past years argued that the jurisprudence that is developed by the courts for Section 9 above is substantive equality kind of judicial precedent. According to De Waal, formal equality refers to treatment without disparities or sameness of treatment", that is, everyone must be treated in the same manner regardless of their circumstance or factors such as historical background, gender, and disability (De Waal 2002, p. 142). Substantive equality, on the other hand, refers to differences in treatment whereby various relevant factors are taken into consideration when different people receive services and opportunities in order to achieve equality (De Waal 2002). In its simplest form, substantive equality entails equity in treatment.

The two forms of equality clearly differ, and only the latter form of equality relates to the transformative constitutionalism agenda that was sought to be achieved by the BEE Amendment Act. This has been confirmed in the case of *President Republic South Africa v*

*Hugo Hugo* case 1997 (6) BCLR 708; 1997 (4) SA 1 (CC), whereby the court affirmed that the interpretation of the right to equality is anchored on substantive equality instead of formal equality. This means the transformative constitutionalism agenda ultimately applies with or through the BEE to achieve the equality objectives (Albertyn and Goldblatt 1998, p. 249). Thus, some groups, such as white people, mostly males, are fairly discriminated against in order to transform the lives of black people, mostly black women.

Transformation through substantive equality requires considering the influence of past events on our current lives. For example, black people inherited poverty and white people inherited wealth from the apartheid regime. This is the reason there are great disparities in wealth in South African societies, and this phenomenon was already in existence long before the Constitution was adopted; hence, the Constitution has a clause that specifically aims to eradicate the inequalities. This means that in the distribution of state resources, people affected by poverty and who were historically disadvantaged should be prioritised, and this is what substantive equality entails. This achieves the greatest form of dignity and equality values, and the BEE Amendment Act aims to achieve these values, although the fronting practices became the impediment towards the BEE goals. The Court has been holding that if a restitutionary measure, even based on any of the grounds of discrimination listed in Section 9(3), passes musters under Section 9(2), it cannot be presumed to be unfairly discriminatory. Hence, substantive equality in the Constitution and various statutes does not only recognise racial equalities but also recognises differences in social and economic standards of living that persist among a wide range of groups. The court, in the case of Minister of Finance v van Heerden, para. 27, reaffirmed that:

> "The Constitution enjoins us to dismantle them and to prevent the creation of new patterns of disadvantage. It is therefore incumbent on courts to scrutinise in each equality claim the situation of the complainants in society; their history and vulnerability; the history, nature and purpose of the discriminatory practice and whether it ameliorates or adds to group disadvantage in real life context, in order to determine its fairness or otherwise in the light of the values of our Constitution."

Furthermore, the Constitutional Court emphasised in the case of National Coalition for Gay and Lesbian Equality v Minister of Justice how the past is often linked with shaping the present and the future and the relevance of substantive equality in mitigating the effects of past negative patterns. The Court explained this as follows:

> "particularly in a country such as South Africa, persons belonging to certain categories have suffered considerable unfair discrimination in the past. It is insufficient for the Constitution merely to ensure, through its Bill of Rights, that statutory provisions which have caused such unfair discrimination in the past are eliminated. Past unfair discrimination frequently has on-going negative consequences, the continuation of which is not halted immediately when the initial causes thereof are eliminated, and unless remedied, may continue for a substantial time and even indefinitely. Like justice, equality delayed is equality denied."

The notion of substantive equality is justified on the ground that the interpretation of equality in every jurisdiction is influenced by the country's historical, social-political, and legal circumstances, and the courts are also using this technique to decide whether the law or policy is unfairly discriminatory (Albertyn 2007, p. 254). Thus, the application and interpretation of the right to equality always necessitate considering relevant factors that should assist courts in achieving equality.

There are, however, various arguments against and in favour of substantive equality. Some arguments, including the contention that the current beneficiaries were not directly affected by the apartheid regime and that the current discriminatory people did not directly enjoy the fruits of the apartheid regime, make substantive equality unfair currently. However, it is also argued that white people indirectly benefited from the apartheid system by inheriting the wealth of their ancestors, which was unfairly accumulated during the

apartheid era (Mushariwa 2011, p. 441). On the other hand, black people inherited poverty from their ancestors since their wealth was unjustifiably taken away from them during the apartheid era, and such a phenomenon still has effects currently, which justifies substantive equality (Ebrahim 2018, p. 6). Section 9 of the Constitution is the bedrock for transformative equality and is used as a yardstick to appraise cases dealing with equality and discrimination. South African courts practically take into consideration numerous factors, such as race, gender, socio-economic background, history, age, socio-political background, status, citizenship, and other related factors, to determine whether equality rights are violated or not. Courts have repeatedly made it clear that in order to logically understand the manner in which equality should be adjudicated, one should look into the nature and sources of inequalities that caused disparities in the past and continue to haunt the present. Courts have in this manner justified the correctness of statutes such as the BEE Act, BEE Amendment Act, and Employment Equity Act 55 of 1998, which make use of affirmative action and thus substantive equality to justify the inequalities or justifiable discriminations that currently transpire in the distribution of socio-economic opportunities.

As the apartheid regime used laws to cause socio-economic inequalities among black people, it is necessary for the current government to also use laws such as the BEE Amendment Act to correct past injustices. The discriminatory statutes that were used by the apartheid regime as tools to deprive black people of their agrarian and residential lands were the Native Land Act of 1913, the Bantu Authorities Act of 1951, and the Native Trust and Land Act of 1936. These are the Acts that created the homelands for black clans in South Africa. Section 1(1)(a) of "the Native Land Act further prohibited black people from purchasing land in all areas where the Native Land Act did not permit residence or land ownership for black people. These Acts perpetuated a long-lasting circle of poverty among black communities since they were forcefully removed from their valuable lands and placed on lands that only allowed them to be employees and produced nothing.

The impact of the above-mentioned statutes resulted in 8% of the South African land being apportioned to black majority people in South Africa to use for residence, agriculture, and anything else they wanted. The Native Land Act created the Bantustans for black people and, thereby, forcefully and without any compensation, removed them from their lands, which were mostly suitable for industrial, farming, and other social-socio economic activities.

The Bantustans (black homelands) were officially abolished by the Abolition of Racially Based Land Measures Act 1991, which repealed all the mentioned discriminatory statutes enacted by the government led by white people during the apartheid era. The only effect the Abolition of Racially Based Land Measures Act had was to put to an end all laws that were racially discriminatory but never attempted to redress the past unlawful land dispossessions and their economic effect on black people. To redress the effect of the apartheid laws correctly and appropriately, the government should technically discriminate in order to achieve equality and balance social-socio-economic differences existing among recognised groups. The South African economy since colonisation has for centuries been systemically sided to cater to white people unfairly economically, socially, and politically and to discriminate against black people.

The Court's duties include, among others, the promotion of equality, which may result in the diminishing of unfair discrimination against marginalised members of the South African Community. This can be seen in the case of Qwelane v South African Human Rights Commission CCT13/20 [2021] ZACC 22 CCT13/20 [2021] ZACC 22, where the Constitutional court's reasons provided guidelines in regard to how to apply substantive equality. The Constitutional Court "provided that the promotion of equality can be achieved by identifying further groups of vulnerable persons and protecting their interests through the constitutional values of equality and dignity.

It is worth noting that although the Constitution, especially the Bill of Rights, aims to achieve equality in every aspect of life for everyone, the equality clause was fundamentally included to address the consequences of apartheid laws. Consequently, the equality clause

is aimed at resolving racial, gender, and any other forms of unfair discrimination that transpired in the past and still affect the present livelihood of the majority population in South Africa. Some of the disadvantages experienced by black people were pointed out by the Constitutional Court in the case of Brink v Kitshoff NO (CCT15/95) [1996] ZACC 9; 1996 (4) SA 197; 1996 (6) BCLR 752, when it pointed out that "Black people were prevented from becoming owners of property or even residing in areas classified as 'white', which constituted nearly 90% of the landmass of South Africa; senior jobs and access to established schools and universities were denied to them; civic amenities, including transport systems, public parks, libraries and many shops were also closed to black people. Instead, separate and inferior facilities were provided". The Constitutional Court also said that apartheid laws have an impact even after their abolishment, stating that the deep scars of this appalling programme are still visible in our society. It is universally accepted that one of the ways to address the consequences of apartheid laws or segregation laws is the imposition of affirmative action in all aspects of life. BEE objectives have affirmative actions that benefit black people in order to eradicate the lingering negative effects of the apartheid regime, which still exist even today (Burger and Jafta 2010, p. 11).

The affirmative action aspects relate to giving priority to certain groups in respect of opportunities and tender-jobs distribution in the public sector. Thus, companies that are owned and managed by black people and women are given preference among all companies when tenders are awarded in the public sector. This is the way the B-BBEE objectives are practically achieved through affirmative action. Therefore, with a clear understanding of substantive equality and the objects of the BBE laws, fronting practices undermine and impede the transformation of social-socioeconomic situations for black people and therefore perpetuate the endless circle of poverty for this group. This means fronting practices are anti-transformative conduct that should not be tolerated in a democratic country where there is still unsatisfactory progress towards realising the social-socioeconomic rights of black people.

## 6. Justifying Anti-Fronting Practices in South Africa

The main reason behind fronting practices is the way South African companies and corporations are affected by the B-BBEE laws, and hence they use fronting techniques to avoid the effects of B-BBEE. Many scholars, legal practitioners, and business leaders have had controversial opinions about the B-BBEE laws (Pooe 2013, p. 638). The section critically provides arguments for and against the B-BBEE laws in South Africa. The analysis is based on the pros (positive) and cons (negative) of rationale or jurisprudence aligned with and behind the B-BBEE laws in South Africa.

### 6.1. Arguments for B-BBEE

It has been argued that the BEE laws have positive implications for the social-socioeconomic conditions of all South Africans and businesses. Pike et al. (2018, p. 7) opine that the BEE laws have an ideal motive, which is to restore the imbalance caused by the apartheid regime, and that would create a state that is based on true equality among all groups. Similarly, Reuben and Bobat (2014, p. 4) postulated that the B-BBEE laws are clearly linked with the transformative constitutionalism notion and substantive equality, although the B-BBEE laws have a special focus on the corporate arena. There is indeed a clear relationship between transformative constitutionalism and BBE laws; hence, the latter derives its motive from the former notion. Basically, all the concepts (B-BBEE, substantive equality, and transformative constitutionalism) seek to achieve one goal in common, namely redressing the issues of inequality in South Africa. So, it sounds so logically thoughtful to connect the BEE laws with the fundamental right to equality in the Constitution. Esser and Dekker (2008, p. 159) expand on the objective of BEE by providing that the South African B-BBEE Act not only aims to correct racial imbalances but also strives to promote social investment and the empowerment of communities. Even though B-BBEE critics view BBBEE as a corrupt scheme, positivists have identified that it is contributing towards

a sense that all citizens belong to one community. Although it does not include the redistribution of wealth and bourgeois lifestyles, it does empower the relevant groups to have opportunities to compete with all the race groups equally. Cant and Wiid (2013, p. 709) "identified that without small businesses, the economy would not prosper. Small and medium enterprises not only contribute towards economic growth but also act as key drivers for empowering previously disadvantaged groups through BBBEE". They do this by bringing previously disadvantaged individuals into the economic mainstream. Through this perspective, it becomes clear that supporters of the BEE lean on substantive equality to justify the appropriateness of the B-BBEE laws. This therefore necessitates considering whether substantive equality suffices to just discrimination stemming from BEE laws.

### 6.2. Arguments against B-BBEE

Some pundits view BEE as neither helpful nor necessary, as they claim that it is an imperfect treatment for an intractable social disease (Skedsvold 1996, p. 34). According to Acemoglu et al. (2007, p. 15), the BBBEE does not seem to have any significant effect on firm investment, labour productivity, or profitability. It was essentially identified that BBBEE, in fact, has a negative impact on investment as well as productivity. Furthermore, Kleynhans and Kruger (2014, p. 5) recognised that BBBEE is impacting businesses negatively because of the high costs associated with BBBEE compliance. Also, BBBEE was contributing to economic strain and an increase in tender corruption; hence, it needs to be restructured. It has been observed that the policy of BEE adds structural costs to the economy, to the detriment of the quality and quantity of goods and services available to the public. Furthermore, the BEE carries substantial costs and is very hard to comply with. This is the reason businesses would go to such lengths to avoid it (Le Roux 2022, p. 16). Mbeki (2009, p. 66) vehemently argues that BEE and its subsidiaries—affirmative action and affirmative procurement—strike the fatal blow against the emergence of black entrepreneurship by creating a small class of unproductive but wealthy black crony capitalists made up of ANC politicians, some retired and others not, who have become strong allies of the economic oligarchy. It is surely a fact that the BEE measures are a compromise and do not please everybody. There are many who argue that affirmative action uses reverse discrimination to solve the problem of discrimination (Gillis et al. 2001, p. 17). This argument is brought up by those who are negatively affected by BEE policies and may try to weigh the BEE down for their own benefit, which is something expected.

Another disadvantage of BEE outlined is that it stigmatises the beneficiaries. Every employee and company from a minority that benefits from BEE policies bears the mark of not being the best choice, but only the best choice from the affirmed group, even if the person was selected for being the best available on the competitive job market. Thus, it cannot be denied that BEE policies drive a wedge between individual self-esteem and economic success. This is because BEE policies have a tendency to provide people from certain minorities with jobs they would not have secured otherwise". But the quality of this job could be compromised in surroundings hostile to the group that the employee is from, and this raises doubts as to whether an affirmed employee is suitably qualified and competent for the job or not. The effects of BEE are also manifested through unfinished projects, tenders that were given to BEE companies, and poor services in the public sector. So, one can further argue that the public also suffers one way or another due to BEE policies when they are applied to corruption.

The arguments against BEE can be compared with the arguments against affirmative action. It was stated that the disadvantage of affirmative action that is currently in circulation is that affirmative action leads to the dropping of standards. Supporters of this view argue that it promotes the hiring of less skilled workers. Employers must choose from the best available employees among minorities instead of simply the best available employees in the market.

### 7. Addressing the Contradictions of Whether B-BBEE Amounts to Discrimination

What should be known is that the BEE interventions seek to create a working environment based on principles of substantive equality (McGregor 2003, p. 423). Substantive equality and BEE laws recognise the inequalities of the past and give preferences to relevant and qualifying groups opportunities to achieve equality, although other groups would have to be discriminated against. This, however, requires that the discrimination be justifiable if one seeks to apply substantive equality. Progressively, the Constitutional Court, in the case of Harksen v Lane NO developed guidelines to determine if a certain conduct constitutes prohibited unfair discrimination, and these guidelines are called 'the two-stage analysis. The guidelines are as follows:

(i)     Firstly, does the differentiation amount to 'discrimination'? If it is on a specified ground, then the discrimination will have been established. If it is not on a specified ground, then whether or not there is discrimination will depend upon whether, objectively, the ground is based on attributes and characteristics that have the potential to impair the fundamental human dignity of persons as human beings or to affect them adversely in a comparably serious manner.

(ii)    If the differentiation amounts to 'discrimination', does it amount to 'unfair discrimination'? If it has been found to have been on a specified ground, then unfairness will be presumed. If on an unspecified ground, unfairness will have to be established by the complainant. The test of unfairness focuses primarily on the impact of the discrimination on the complainant and others in his or her situation.

These guidelines were applied with approval in the case of Louw v Golden Arrow Bus Service, whereby the court summarised the application of the guidelines as follows:

(a)     Does the act or omission constitute differentiation between people or categories of people?

(b)     If the answer is positive, the court embarks on a two-stage analysis.

In the case of *IMATU*, the court further simplified the guidelines by reducing them into a test that a court may employ to determine if certain discrimination constitutes prohibited unfair discrimination. The court developed the test by stating that the test of whether a ground is analogous to the listed grounds was whether the alleged discrimination had the potential to impair the fundamental human dignity of persons as human beings or to affect them adversely in a comparably serious manner. It should be noted that "constitutional rights are interdependent and interconnected" (Pieterse 2007, p. 797). Liebenberg and Goldblatt propose what they call an interpretive interdependence of rights (Liebenberg and Goldblatt 2007, p. 337). To this end, they envision a form of interpretive dependence whereby courts are encouraged to consider how values that underlie one right may be of assistance in developing jurisprudence. They proposed that since equality is a foundational value in the Constitution, it must inform the interpretation of all the rights in the Bill of Rights. With regard to equality and socio-economic rights, they note that an approach to the interpretation of equality and social-socioeconomic rights that acknowledges the interrelationship between these rights is also more likely to be responsive to the reality that the most severe forms of disadvantage are usually experienced as a result of an intersection between group-based forms of discrimination and social-socioeconomic marginalisation. The contention is that "such an approach may be of assistance in dealing with the legacies of colonialism and eradicating the effects of apartheid and sexism from stereotypic perspectives, as purported by the BEE laws. Therefore, the BEE interventions do not unfairly discriminate against any person.

### 8. How to Achieve the (BBBEEA 2023)

This paper considers fronting as the major impediment to the realisation of the BBEE. Thus, fronting should be highly tackled while not neglecting other mechanisms to achieve the BBEE. The need to tackle down-fronting was recognised by the 2013 amendments to the B-BBEE Act. The 2013 amendments to the B-BBEE Act entrench mechanisms such as

cancellation of contracts, establishment of the BBEE Commission with its related functions, and broadening the scope of fronting as a criminal offence.

The first mechanism to assist in achieving the BBEE is entrenched in the 2013 B-BBEE Amendment Act, which introduces cancellation of contracts where fronting is found to have induced the awarding of tenders. The B-BBEE Amendment Act, in terms of Section 13A, states that any contract or authorization awarded on account of false information knowingly furnished by or on behalf of an enterprise in respect of its broad-based black economic empowerment status may be cancelled by the organ of state or public entity without prejudice to any other remedies that the organ of state or public entity may have. Parties often had to rely on courts to cancel contracts, but this amendment allows for the cancellation of contracts where fronting is found to have taken place. This can assist in achieving the BBEE; hence, state entities are empowered to make use of quick and cost-effective methods of cancelling contracts that are mischievous to achieving the BBEE.

Secondly, the 2013 Amendment Act brought about the establishment of the BBEE commission. In addition, the BBEE Commission is mandated to perform functions that are aimed at achieving BBEE. The functions of the BBEE Commission include: (a) to oversee, supervise, and promote adherence to this Act in the interest of the public; (b) to strengthen and foster collaboration between the public and private sectors in order to promote and safeguard the objectives of broad-based black economic empowerment; (c) to receive complaints relating to broad-based black economic empowerment in accordance with the provisions of this Act; and (d) to investigate, either on its own initiative or in response to complaints received, any matter concerning broad-based black economic empowerment.

In addition to the above functions, the BBEE commission has powers to, on its own initiative or on receipt of a complaint in the prescribed form, investigate any matter arising from the application of the Act, including any B-BBEE initiative or category of B-BBEE initiatives.[1] This means that the BBEE Commission will receive complaints and initiate its own investigation. This can assist in instances where fronting occurred and was concealed by government officials. Under these circumstances, government officials cannot be reasonably expected to conduct investigations, which will lead to the uncovering of fronting and corruption in awarding contracts. Therefore, the BBEE Commission becomes even more relevant to combating fronting and corruption since it will conduct impartial investigations.

The last mechanism is broadening the scope of the offence. Section 130(1) of the B-BBEE Amendment Act, 2013 states that a person commits an offence if that person knowingly: (a) misrepresents or attempts to misrepresent the broad-based black economic empowerment status of an enterprise; (b) provides false information or misrepresents information to a B-BBEE verification professional in order to secure a particular broad-based black economic empowerment status or any benefit associated with the compliance with this Act; (c) provides false information or misrepresents information relevant to assessing the broad-based black economic empowerment status of an enterprise to any organ of state or public entity; and (d) engages in a fronting practice.

The above provisions seek to combat fronting by not only penalising business owners that engage in fronting but also those who intentionally provide false information aiming to help a certain enterprise achieve a certain BBEE level. Thus, this means any person who falls within the ambit of Section 130(1) will be liable for the punishment outlined in Section 130(3)(a) of the BEE Amendment Act. The latter provision provides that an individual who knowingly engages in a fronting practice will be liable to a fine and/or imprisonment for a period not exceeding 10 (ten) years. When applied together, these provisions seek to deter business owners and people falling within BBEE groups from refraining from fronting practices.

There are a limited number of cases where fronting was prosecuted as an offence. One of the cases is the case of S v Avril Elizabeth Home for the Mentally Handicapped (2012) ZAGPPHC 107, whereby the court found that Avril Elizabeth Home for the Mentally

---

[1] Section 13J of the B-BBEE Amendment Act, 2013.



Handicapped had engaged in fronting by falsely claiming to be a black-owned company in order to obtain a B-BBEE certificate. The court ordered the company to pay a fine of ZAR 100,000 (approximately USD 7000) and to forfeit the B-BBEE certificate.

Another case is the case of S v Siyenza Group 2017 (2) SACR 1 (GP), whereby the court found that Siyenza Group had engaged in fronting by using a black-owned company as a front to obtain a B-BBEE certificate while retaining effective control of the company. The court ordered Siyenza Group to pay a fine of ZAR 1.5 million (approximately USD 105,000) and to forfeit the B-BBEE certificate.

## 9. Concluding Remarks

The journal paper is informed by the increasing number of fronting reports recently published by the Department of Trade, Industry, and Competition. Thus, the paper sought to analyse the post-1994 anti-fronting mechanisms to foster broad-based economic inclusion of black people to participate in the mainstream of the economy. The paper noted that the right to equality is the core foundation of the BEE laws in South Africa. The equality right is founded in Article 7 of the UDHR and entrenched in Section 9 of the Constitution. The right to equality takes the form of substantive equality, and that is how it is linked with the BEE laws. The BEE Act and BEE Amendment Act both seek to transform the socio-economic conditions of black people through substantive equality or transformative constitutionalism schemes. However, the fronting conducts undermine the progress towards achieving this objective. However, the BEE Amendment Act criminalises fronting, and that gives better hope that many people will be deterred from fronting activities. The advantages and disadvantages can perhaps be summed up thus: despite all the negative aspects of BEE policies, they are necessary. Until a better solution is found, BEE policies should stay, being the best solution available as a compromise. Based on this, it is impossible to implement ideas to eradicate the consequences of apartheid without an objection from the minority group of people (including a small number of black people) in South Africa. It is also found that the B-BBEE Amendment Act of 2013 entrenches various anti-fronting mechanisms, such as widening the scope of fronting offences, reports, cancellation of contracts, the establishment of the BBEE Commission, and introducing independent functions of the BBEE Commission. Therefore, the paper advises that in order to effectively put into practice pro-active anti-fronting measures, the BBEE Commission should be given the role of receiving audit reports and initiating its own investigation where necessary to combat the growing fronting practices. If these anti-fronting measures are implemented properly, they can foster the inclusion of black people into mainstream economic participation in South Africa. The paper adds that individuals found to have committed fronting personally or through their companies should be given harsher punishments, such as long direct imprisonment sentences with no option of a fine. This can cause financial distress to the perpetrators and serve as reactive mechanisms.

**Author Contributions:** Conceptualization, T.H.M. and K.O.O.; writing—original draft preparation, T.H.M. and K.O.O.; writing—review and editing, T.H.M. and K.O.O. All authors have read and agreed to the published version of the manuscript.

**Funding:** Their research received no external funding.

**Institutional Review Board Statement:** Not applicable.

**Informed Consent Statement:** Not applicable.

**Data Availability Statement:** Not applicable.

**Conflicts of Interest:** The authors declare no conflict of interest.

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
