# Peer review of "An Analysis of Post-Apartheid Anti-Fronting Interventions Fostering Mainstreaming of the Black South Africans into Corporate Sector"

_laws_

Round 1
Reviewer 1 Report
Comments and Suggestions for Authors
Line 77-84 is better expressed by the author(s) in a prose form rather than bullet form
Line 85-86 'the white in cohort with some undesirable blacks have continually' These word should be reviewed, unless the author(s) is quoting a source and the author(s) must be able to tell his audience that it cannot be all the white and undesirable should be replaced with a gentler word. the mere fact that a person's view does not correspond with the majority's view does not make him/her undesirable.
Line 351-352-the sentence is not correct grammatically. One word must be removed. Revisit
Line 384- the sentence is not correct grammatically. Revisit
Comments on the Quality of English LanguageThe author(s) grasp of English language is excellent, except for a few pardonable errors highlighted above, the quality of the language employed is adequate.
Reviewer 2 Report
Comments and Suggestions for Authors
Thank you for the opportunity to read this fascinating and informative piece around the policy interventions in South Africa to address inequality and the current shortcomings of the interventions. The article introduces the historical background to the broad socio-economic imbalances in South Africa, explains the various policy interventions meant to address inequality, the legal context for these policies, the case law supporting the relevance of substantive equality, along with the debates around these policy interventions. The piece identifies the practice of fronting that undermines the policy interventions and circumvents its implementation. The policy is presented through the lens of the principle of substantive equality as enshrined in the Constitution and upheld through case law. The article is rich with detail and analysis of the various legal and policy instruments, which are contextualized as key features of the post-1994 government.
The piece could be improved with a clearer presentation of the argument and better organization of the sections. As it stands, the mentions of fronting and anti-fronting are few and not well integrated into the paper. To strengthen the paper, I would suggest the following reorganization:
1. Add a few paragraphs of introduction to introduce fronting as an impediment to implementing progressive policies. What will this article tell us about fronting and the ways in which South African law is/is not able to address this impediment to substantive equality? What has previous literature shown about the topic of fronting?
2. Then move into the historical background, explaining why anti-fronting as a policy was important for SA in the post-1994 period based on the principle of substantive equality enshrined in the Constitution. Link anti-fronting to this principle and the subsequent case law you discuss.
3. Then introduce the different policies, the BEEE and the B-BBEE Act, and the subsequent debate around these laws. Link them closely to fronting. Do weaknesses in the law help explain the persistence/impunity around fronting?
4. The extent of fronting in SA is not made clear. Perhaps some data could be provided on the extent to which the practice is known or not known to exist. Is this a common practice in other countries as well?
5. Then introduce the efforts to address fronting by the government.
6. Conclude as you do by reminding how fronting undermines the principles as laid out in the Constitution and the policy interventions that intend to address inequality. What new information has your article provided? For example, can you make suggestions for how BEE can policies be improved to resist fronting practices? What more needs to happen?
I hope these suggestions offer a road map for elevating your important arguments and insights.
Comments on the Quality of English Language
Moderate editing of English language required
Reviewer 3 Report
Comments and Suggestions for Authors
Dear author,
The article has an original approach.
The object of the article needs to be well defined. The subject is relevant.
Need to develop the methodology better. In terms of theoretical approach, it brings current and relevant articles.
I understand that there should be minor revisions:
1) In the abstract: express the objective, methodology and main results of the research.
2) In the introduction: you need to write the research problem, objective and methodology clearer.
3) Need to better develop the conclusion, and relate it to the research objective.
4) You need to provide more data (empirical research) about your propositions.
Comments on the Quality of English LanguageThe text only needs a minor revision.
Round 2
Reviewer 3 Report
Comments and Suggestions for Authors
The article is ready for publication.
Best regards,